# Diversity of the Genomes and Neurotoxins of Strains of *Clostridium botulinum* Group I and *Clostridium sporogenes* Associated with Foodborne, Infant and Wound Botulism

**DOI:** 10.3390/toxins12090586

**Published:** 2020-09-11

**Authors:** Jason Brunt, Arnoud H. M. van Vliet, Andrew T. Carter, Sandra C. Stringer, Corinne Amar, Kathie A. Grant, Gauri Godbole, Michael W. Peck

**Affiliations:** 1Department of Chemical Engineering and Biotechnology, University of Cambridge, Philippa Fawcett Drive, Cambridge CB3 0AS, UK; 2Gut Health and Food Safety, Quadram Institute, Norwich Research Park, Norwich NR4 7UQ, UK; andrewcarter594@btinternet.com (A.T.C.); sandra.stringer@quadram.ac.uk (S.C.S.); 3School of Veterinary Medicine, Faculty of Health and Medical Sciences, University of Surrey, Guildford GU2 7AL, UK; a.vanvliet@surrey.ac.uk; 4Gastrointestinal Pathogens Unit, National Infection Service, Public Health England, London NW9 5EQ, UK; Corinne.Amar@phe.gov.uk (C.A.); kanngrant1@gmail.com (K.A.G.); Gauri.Godbole@phe.gov.uk (G.G.)

**Keywords:** *Clostridium botulinum*, Botulism, Neurotoxins, *Clostridium sporogenes*, Genomes, Foodborne, Infant, Wound

## Abstract

*Clostridium botulinum* Group I and *Clostridium sporogenes* are closely related bacteria responsible for foodborne, infant and wound botulism. A comparative genomic study with 556 highly diverse strains of *C. botulinum* Group I and *C. sporogenes* (including 417 newly sequenced strains) has been carried out to characterise the genetic diversity and spread of these bacteria and their neurotoxin genes. Core genome single-nucleotide polymorphism (SNP) analysis revealed two major lineages; *C. botulinum* Group I (most strains possessed botulinum neurotoxin gene(s) of types A, B and/or F) and *C. sporogenes* (some strains possessed a type B botulinum neurotoxin gene). Both lineages contained strains responsible for foodborne, infant and wound botulism. A new *C. sporogenes* cluster was identified that included five strains with a gene encoding botulinum neurotoxin sub-type B1. There was significant evidence of horizontal transfer of botulinum neurotoxin genes between distantly related bacteria. Population structure/diversity have been characterised, and novel associations discovered between whole genome lineage, botulinum neurotoxin sub-type variant, epidemiological links to foodborne, infant and wound botulism, and geographic origin. The impact of genomic and physiological variability on the botulism risk has been assessed. The genome sequences are a valuable resource for future research (e.g., pathogen biology, evolution of *C. botulinum* and its neurotoxin genes, improved pathogen detection and discrimination), and support enhanced risk assessments and the prevention of botulism.

## 1. Introduction

*Clostridium botulinum* Group I (also known as proteolytic *C. botulinum*) and *Clostridium sporogenes* are closely related mesophilic bacteria that share genotypic and physiological characteristics, including a highly proteolytic nature and the ability to form spores of high thermal resistance. *C. botulinum* Group I is one of four distinct groups within the species *C. botulinum*, a polyphyletic taxon united only by the ability to form the highly potent botulinum neurotoxin [1,2,3,4,5]. *C. sporogenes* has often been perceived as a non-toxic close relative of *C. botulinum* Group I [6,7,8,9,10,11,12,13]. It is an important cause of food spoilage [14,15,16], and used as a surrogate for *C. botulinum* Group I in food sterilisation tests [7,14,15,17,18,19,20]. Recent findings, however, show that *C. sporogenes* is not merely a non-toxic form of *C. botulinum* Group I [8,10,21,22,23,24,25]. Genetic analysis has revealed that both *C. botulinum* Group I and *C. sporogenes* contain strains that either form or do not form botulinum neurotoxin. Most strains of *C. botulinum* Group I (but not all) form botulinum neurotoxin, whilst some (but not most) *C. sporogenes* strains also form botulinum neurotoxin [8,10,22,24,25]. If *C. botulinum* Group I and *C. sporogenes* are considered separate species, then *C. sporogenes* represents another distinct bacterium that forms botulinum neurotoxin [22,24,25,26].

Strains of *C. botulinum* Group I (and to a lesser extent *C. sporogenes*) are a major cause of the three most frequent types of botulism in humans (foodborne, infant and wound botulism) and are also responsible for botulism in animals [4,5,27,28,29,30]. Foodborne botulism is a severe, and often lethal, neuroparalytic intoxication potentially caused by the consumption of as little as 50 ng of botulinum neurotoxin [5,31]. Spores of *C. botulinum* Group I are the target of the Botulinum cook (121 °C/3 min) given to low acid canned foods [4,5]. Foodborne botulism outbreaks have been associated with a failure to apply the Botulinum cook to canned or bottled foods, and also with temperature abuse of products intended to be stored chilled [3,5,32,33,34,35,36]. The commercial implications of foodborne botulism outbreaks can be significant [4,5], and continued extreme vigilance is important to ensure that the incidence of foodborne botulism is minimised. Infant and wound botulism are both infections in which the bacterium multiplies and forms botulinum neurotoxin in the body. Infant botulism results from the absorption of botulinum neurotoxin formed by the bacterium following colonization of the intestinal tract of infants under twelve months. Honey has been identified as one source of the bacterial spores. Wound botulism ensues following the formation of botulinum neurotoxin by the bacterium multiplying in a wound. Initial cases were associated with traumatic injury, but over the last few decades there has been an increase in cases associated with injectable drug abuse [5,27,37,38].

Botulinum neurotoxins are zinc metalloproteases that ultimately reach the nerve cell cytoplasm where they specifically cleave SNARE proteins involved in release of the neurotransmitter, acetylcholine, resulting in a severe and often deadly flaccid muscle paralysis known as botulism [39,40]. More than a century ago, two distinct botulinum neurotoxin serotypes were identified using specific antisera to neutralise botulinum neurotoxin in animal tests [27,41,42]. Burke referred to these as serotypes A and B [41]. Today, many botulinum neurotoxin serotypes and more than 40 botulinum neurotoxin sub-types are recognised [43,44,45,46]. A majority of botulinum neurotoxin sub-types have been identified through the sequencing of genes encoding botulinum neurotoxins, and the derivation of the amino acid sequence. Most sub-types differ by at least 2.6% in amino acid sequence from another sub-type [45,46], and sub-types of botulinum neurotoxins serotypes A and B have been shown to exhibit distinct functional and toxicological properties [47,48,49,50,51]. Each botulinum neurotoxin sub-type comprises a series of variants with unique amino acid sequence.

The strains examined in the present study possess up to three neurotoxin genes, and form up to three neurotoxins of type A, B and/or F. Strains with two neurotoxin genes form either one active toxin (e.g., type A(B) strains possess a type A and type B neurotoxin gene, but only form type A neurotoxin) or two active toxins (e.g., type Bf strains form a greater amount of type B neurotoxin than type F neurotoxin). In the present study, strains are referred to by their type or sub-type of botulinum neurotoxin gene(s), e.g., type A strains possess a gene encoding botulinum neurotoxin type A, B5F2 strains possess genes encoding botulinum neurotoxin sub-types B5 and F2. The botulinum neurotoxin is accompanied by accessory proteins (notably, non-toxic-non-haemagglutinin (NTNH)) in a series of toxin complexes that are encoded by genes located in either a *ha* cluster or an *orfX* cluster. The *ha* cluster includes genes encoding the neurotoxin (all type B neurotoxins and some type A neurotoxins), NTNH, three haemagglutinins, and a positive regulator. The *orfX* cluster includes genes encoding the neurotoxin (all type F neurotoxins and other type A neurotoxins), NTNH, a positive regulator, and four open reading frames of unknown function [3,43,44,45,46]. The neurotoxin gene(s) in *C. botulinum* Group I and *C. sporogenes* can be located on the chromosome or a plasmid.

The purpose of the present study was to provide insights into the genetic diversity and spread of strains of *C. botulinum* Group I and *C. sporogenes* and their neurotoxin genes. The genomes of 417 newly sequenced highly diverse strains (isolated from 24 countries and six continents over a period of more than one hundred years) has been supplemented with 139 genomes publicly available in Genbank. This represents the most comprehensive comparative genomic study of these bacteria currently undertaken. Furthermore, the 556 genome sequences also represent a valuable resource, contributing to future genomic, transcriptomic, proteomic and systems biology endeavours such as dissecting the evolution of *C. botulinum* Group I and *C. sporogenes* and their neurotoxin genes, the analysis of spore properties, regulation of neurotoxin formation, and other characteristics important for pathogen transmission, and also the targeting of PCR assays to detect, trace and discriminate strains [15,17,52,53,54,55,56]. These findings may add to future risk assessments and to the prevention of botulism in humans and animals.

## 2. Results and Discussion

### 2.1. Overview of Diversity of Genomes and Neurotoxins

Publication of genome sequences of *C. botulinum* Group I began with that of strain ATCC 3502 (Hall A strain 174) [57], and at the start of this work there were 139 genomes publicly available in Genbank. This sequence information has accelerated studies into the diversity and phylogenetic relationship of strains of *C. botulinum* Group I and *C. sporogenes* [5,22,24,25,29,46,57,58,59,60,61,62,63]. Sequencing of botulinum neurotoxin genes/clusters has provided insight into the diversity and evolution of encoded neurotoxin serotypes, sub-types and variants [45]. In the present study, the genomic diversity, phylogenetic relationship and botulinum neurotoxin genes/clusters have been characterised for 556 strains of *C. botulinum* Group I and *C. sporogenes* (Appendix A). This includes 417 newly sequenced strains, and considerably increases the number of publicly available genome sequences for *C. botulinum* Group I and *C. sporogenes*. The strains were from a wide geographical area (24 countries from Europe (predominantly the UK), Asia, Africa, Australasia, north America and south America; Figure 1). Seventy-three strains were associated with foodborne botulism, 70 strains with infant botulism, and 82 strains with wound botulism (Appendix A). A further 139 strains were non-clinical isolates not associated with botulism in humans, and the status of the remaining isolates is unknown.

This study has included a number of isolates associated with UK cases of foodborne, infant and wound botulism, and also some UK non-clinical isolates (Appendix A). Botulism incidents reported in the UK between 1989 and 2019 are summarized in Table 1, with details of initial cases described previously [64,65,66]. Recent cases of foodborne botulism have often been associated with the consumption of imported food and have been associated with strains possessing genes encoding botulinum toxin sub-types A1, B1, B2 and B5F2 (Table 1). Sporadic cases of infant botulism continue to be recorded and are usually associated with known risk factors such as weaning, contact with dusty environment during travel and feeding honey to the infant. Strains possessing genes encoding botulinum toxin sub-types A2, B2, B5 and B5F2 have been associated with infant botulism between 1989 and 2019 (Table 1). There have been several outbreaks of wound botulism related to injectable drug use, with the first UK case reported in 2000. Several wound botulism outbreaks often involving multiple patients are indicated in some years. The most recent outbreak of wound botulism in Scotland in injectable drug users occurred due to a large contaminated batch of heroin [67]. A1B5, A5B2, A5B3 and B3 strains have been associated with wound botulism (Table 1). A single case of adult colonization botulism was recorded in 2018 (Table 1).

The 556 strains were assigned to one of two major lineages (*C. botulinum* Group I or C. *sporogenes*) based on an analysis of core genome single nucleotide polymorphisms (SNPs) following whole genome sequencing, and in silico multi-locus sequence typing (MLST) following the scheme of Jacobson et al. [68]. Each lineage comprised a series of distinct clusters (Figure 2). This is consistent with observations made previously that have largely been carried out on a smaller number of and/or less diverse strains [3,5,7,8,22,24,25,29,43,44,46,58,60,62,68]. The genome of 95% (429 out of 452) of strains assigned to the *C. botulinum* Group I lineage possessed a botulinum neurotoxin gene or genes, while the genome of only 19% (20 out of 104) of strains assigned to the *C. sporogenes* lineage possessed a botulinum neurotoxin gene. This assignation resulted in a few strains now being designated as *C. botulinum* Group I and not *C. sporogenes*, or vice-versa. For example, strains 2345, AM370, and B2 450 were all previously described as *C. botulinum* Group I on the basis of their ability to form botulinum neurotoxin [8,25,29,59], but in the present study are located within the *C. sporogenes* lineage as their genome closely aligns with that of other strains of *C. sporogenes*.

Clusters of strains can be identified in the two lineages, with most clusters dominated by strains possessing genes that encode the same botulinum neurotoxin sub-type(s). There are also, however, examples of genes encoding the same botulinum neurotoxin sub-type(s) being located in different genomic backgrounds, indicating frequent horizontal movement of botulinum neurotoxin genes between clusters (Figure 2). A majority of genomes (354) possessed a single gene encoding a botulinum neurotoxin (type A, 183 strains; type B, 154 strains; type F, 17 strains), while 89 strains possessed two genes encoding botulinum neurotoxins, and the genome of six strains included three genes encoding botulinum neurotoxins. The present study did not determine whether the presence of a neurotoxin gene was associated with formation of botulinum neurotoxin. The genomes of 107 strains did not possess a gene encoding botulinum neurotoxin. The genomes of all strains were searched using an approach previously used to discover a novel putative botulinum neurotoxin gene in *Enterococcus* [69], but this failed to reveal genes that might encode previously undescribed botulinum neurotoxin serotypes or sub-types. Novel botulinum neurotoxin sub-type variants were identified, each with unique coding (and predicted amino acid) sequence.

**Figure 2 toxins-12-00586-f002:**
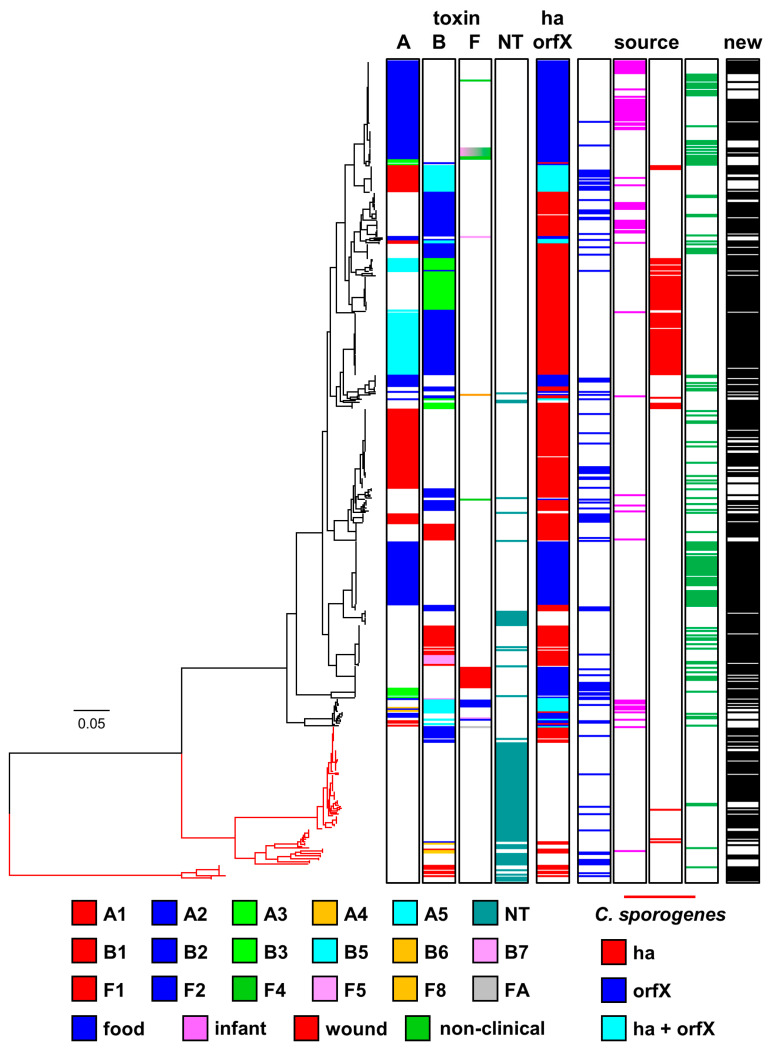
Phylogeny of genomes of *C. botulinum* Group I and *C. sporogenes*. Two major lineages (*C. botulinum* Group I (shown in black) and *C. sporogenes* (shown in red)) were identified. The phylogenetic tree was created by comparison of core single nucleotide polymorphisms identified using ParSNP program [70], Treegraph v2 [71], MEGA7 [72] and Figtree were used to annotate and visualize the phylogenetic tree. Accessory genes were identified using BLAST using reference sequences for the *ha* and *orfX* neurotoxin cluster configurations. The distance bar (0.05) represents the number of nucleotide substitutions per site for a given branch, based on the number of SNPs found in the core genome. Details of individual sequences and strains are given in Appendix A, where they appear in the same order as in this Figure. Newly sequenced strains are indicated (in black) as “new” in the right-hand column.

### 2.2. Diversity and Phylogeny of C. botulinum Group I type A Strains and Their Botulinum Neurotoxin Genes

Eight sub-types of botulinum neurotoxin type A (A1 to A8) are currently recognised. These sub-types have an amino acid difference of 2.9–15.6%, and within sub-type differences of 0.1–2.5% [45]. The genome of 271 strains examined in the present study contained a type A neurotoxin gene. For 183 strains, the type A neurotoxin gene was the only neurotoxin gene present, with 88 strains containing an additional neurotoxin gene(s) (Figure 2). Type A neurotoxin genes were located in strains within the *C. botulinum* Group I lineage, but not the *C. sporogenes* lineage (Figure 2). Strains possessing a gene encoding the same type A neurotoxin sub-type were located together in a series of discrete clusters. However, strains within these individual clusters were often more closely related to strains with other neurotoxin genes or no neurotoxin gene, than to strains in other clusters with the same sub-type A neurotoxin gene (Figure 2). The presence of genes encoding the same botulinum neurotoxin sub-type A gene in different genomic backgrounds indicates horizontal transfer of type A botulinum neurotoxin genes between clusters.

The genome of 84 strains examined in the present study encoded a gene for botulinum neurotoxin sub-type A1 (Figure 2). Genes encoding sub-type A1 neurotoxin were the only botulinum neurotoxin genes located in either a *ha* or *orfX* neurotoxin gene cluster. Three major variant groupings of botulinum neurotoxin sub-type A1 were recognised, and these were encoded by A1B5 strains, A1 (*ha*) strains and A1 (*orfX*) strains (Figure 3, Appendix A). This is as reported previously [3,22,45], with no new major variant grouping identified. Core genome SNP analysis separated strains with a sub-type A1 neurotoxin gene into five clusters within *C. botulinum* Group I. Two clusters contained A1B5 strains, two clusters contained A1 (*ha*) strains, while all A1 (*orfX*) strains were located in a single cluster (Figure 2). A1B5 strains possessed a genome that encoded a sub-type A1 neurotoxin (central sub-type A1 variant group in Figure 3 and Appendix A) located in an *orfX* neurotoxin gene cluster, and also a putative sub-type B5 neurotoxin. Eighteen strains belonged to one cluster and MLST ST-4 or ST-6, while two strains belonged to the second cluster and MLST ST-10 (Figure 2, Appendix A). Previous studies found that many other A1B5 strains also belonged to ST-4 [68,73]. A1B5 strains were associated with foodborne botulism in Ireland (in 2006) and USA, infant botulism in UK (in 1978) and USA, and wound botulism in UK (in 2009) (Table 1 and Appendix A) [65].

Sixty-three A1 (*ha*) strains possessed a single gene encoding botulinum neurotoxin belonging to the upper sub-type A1 variant group in Figure 3 and Appendix A (in a *ha* neurotoxin gene cluster). These strains were separated into two distinct clusters by core genome SNP analysis (Figure 2). A majority of strains (54) belonged to one cluster and MLST ST-1 or ST-9, while a minority of strains (seven) were present in a second cluster and MLST ST-19 (Figure 2, Appendix A). Strains were associated with foodborne botulism, including the 2011 UK outbreak involving three children who consumed korma sauce (Table 1) [74], and various incidents in the USA, but not with wound or infant botulism. Non-clinical isolates with genes encoding this variant were from Sweden, Kazakhstan, USA and Netherlands (Appendix A). Mazuet et al. [29] reported on a number of similar sub-type A1 (*ha*) strains that belonged to MLST ST-1 or ST-19 and were associated with foodborne botulism or animal botulism in France. The fifth cluster comprised three closely related strains (Figure 2), with a gene encoding sub-type A1 neurotoxin (lower sub-type A1 variant in Figure 3 and Appendix A) in an *orfX* neurotoxin cluster (A1 (*orfX*) strains). Two of the three strains were associated with foodborne botulism, and the third is a non-clinical isolate (Figure 2). The present study concurred with previous reports that A1 (*orfX*) strains were more closely related to strains of other sub-types (e.g., A4B5, B5F2) than to other sub-type A1 strains [8,25,60,75,76].

A gene encoding sub-type A2 neurotoxin was present in 128 strains (Figure 3), some of which were associated with foodborne or infant botulism, but not wound botulism (Figure 2 and Figure 3). All sub-type A2 neurotoxin genes were located in an *orfX* neurotoxin gene cluster. Some strains contained additional type B or type F neurotoxin gene(s) (Figure 2). Sub-type A2 strains separated into six clusters within *C. botulinum* Group I (Figure 2). A majority of strains shared a common genomic background (upper cluster in Figure 2), and a gene encoding the most frequently recorded sub-type A2 botulinum neurotoxin variant (upper variant group in Figure 3, Appendix A). These strains belonged to MLST ST-2, ST-22 and ST-26 (Appendix A). Sub-type A2 strains examined in the present and a previous study principally belonged to ST-2 and ST-26 [77]. However, an additional fifteen strains examined in the present study with more distant genomic backgrounds according to core genome SNP analysis and MLST analysis (ST-7, ST-28, ST-38, ST-61, ST-85) (Figure 2, Appendix A) also possessed a gene encoding this sub-type A2 neurotoxin variant. The gene encoding this sub-type A2 neurotoxin variant has therefore moved between distantly related *C. botulinum* Group I strains by horizontal gene transfer. Strains with this sub-type A2 variant gene were associated with foodborne botulism, all of the infant botulism cases associated with sub-type A2 strains, and there were also non-clinical isolates. This included strain Mauritius that was associated with an outbreak of foodborne botulism in the UK in 1955 following consumption of pickled fish prepared in the Indian Ocean island of Mauritius [66], and strains responsible for five cases of infant botulism in the UK (Table 1) [65,78], and infant botulism in Japan and USA [79]. Non-clinical isolates included strains from Argentina [23,25], and two strains isolated in our laboratory in 2015 from soil samples from Uganda (Stringer SC and Peck MW, unpublished results). Strain CDC 53174 that was isolated from stool sample in Uganda and our two sub-type A2 strains from Uganda (IFR 15/031 and 16/622) all belong to ST-28 [77]. The first reported outbreak of foodborne botulism in Uganda involved a strain of *C. botulinum* Group I type A [80]. Few strains of *C. botulinum* have been isolated from Africa, and it is noteworthy that the genomic background and neurotoxin sub-type variant align with that found in other strains from around the world.

All but one of the 44 strains with a genome encoding the second most frequently recorded sub-type A2 botulinum neurotoxin variant (second variant group in Figure 3, Appendix A) shared a common genomic background, as revealed by core genome SNP analysis (Figure 2). The strains were isolated from several countries including UK, Denmark, Norway, Canada and Japan, with none known to be associated with botulism (Figure 2, Appendix A). Single, distantly related, strains possessed genomes encoding each of the other two sub-type A2 botulinum neurotoxin variants. A strain associated with an outbreak of foodborne botulism involving mascarpone cheese in Italy in 1996 [59] encoded the sole example of the third sub-type A2 variant (third variant group in Figure 3, Appendix A). A strain associated with an outbreak of foodborne botulism involving home-canned meat in Italy in 1993 [59] provided the only example of the fourth sub-type A2 botulinum neurotoxin variant (fourth variant group in Figure 3, Appendix A). These strains appeared to be more closely related to strains with other botulinum neurotoxin genes, than to each other or to other strains that possessed a gene encoding botulinum sub-type A2 neurotoxin (Figure 2).

Core genome SNP analysis and MLST analysis showed that strains with a gene encoding botulinum neurotoxin sub-type A3 were present in two distinct *C. botulinum* Group I clusters (Figure 2). Strains in the upper cluster (Figure 2) were non-clinical isolates from Argentina [23,25]. These strains belonged to MLST ST-2 or ST-25 (Appendix A), and possessed a gene encoding the lower variant of botulinum neurotoxin sub-type A3 in Figure 3 (Appendix A). Three further sub-type A3 strains from Argentina also belonged to ST-25 [77]. Strains in the lower cluster (Figure 2) included isolates derived from the first reported foodborne botulism outbreak in the UK at Loch Maree in 1922 [66,81], and also included a UK non-clinical isolate. These strains belonged to MLST ST-3 (Appendix A), and possessed a gene encoding a second variant of botulinum neurotoxin sub-type A3 (upper variant in Figure 3, Appendix A). Strains within each of these two sub-type A3 genomic clusters were more closely related to strains with other botulinum neurotoxin genes, than to strains in the other sub-type A3 genomic cluster. Botulinum neurotoxin sub-types A3 and A4 were encoded by genes located in an *orfX* neurotoxin gene cluster. Two closely related strains, as determined by core genome SNP analysis and MLST analysis (ST-7) possessed a genome encoding botulinum neurotoxins sub-type A4 and sub-type B5 (Figure 2). Both strains were associated with infant botulism; strain 657 in Texas in 1976 [82], and strain SRR8527662 in Ireland in 2014 (Appendix A). A single variant of botulinum neurotoxin sub-type A4 was identified (Figure 3).

Wound botulism was first reported in the UK in 2000. Cases of wound botulism reported between 2000 and 2004 were mainly associated with isolates forming type A neurotoxin, with a few isolates forming type B neurotoxin [64,83]. A detailed analysis of isolates implicated in UK wound botulism cases between 2000 and 2019 has shown that sub-type A5 strains were responsible for a number of time-separated wound botulism outbreaks (Table 1). All UK cases of wound botulism were associated with injectable drug abuse, with the drugs and associated strains believed to originate in Asia [8,83]. Genes encoding the sub-type A5 neurotoxin and a truncated sub-type B2 or B3 neurotoxin were co-located in a *ha* neurotoxin gene cluster, as described previously [8,29,84,85,86]. The truncated B2 and B3 genes were of similar length. Two variants of sub-type A5 neurotoxin were identified (Figure 3). Strains either contained genes encoding sub-type A5 neurotoxin belonging to the upper variant group in Figure 3 and a truncated sub-type B2 neurotoxin, or sub-type A5 neurotoxin belonging to the lower variant group in Figure 3 and a truncated B3 neurotoxin. The A5B2 strains were associated with wound botulism cases in the UK between 2004 and 2013 (Table 1), wound botulism in Switzerland in 2000, and foodborne botulism in France in 2011 [29]. Core genome SNP analysis and MLST analysis established that most A5B2 strains (including all associated with wound botulism) were closely related, and belonged to MLST ST-47 (Figure 2, Appendix A). Two A5B2 strains (including one associated with foodborne botulism) were more distantly related according to core genome SNP analysis and belonged to different MLST sequence types (ST-46 and ST-86). The A5B3 strains were associated with wound botulism cases in the UK between 2000 and 2012, and were closely related, based on core genome SNP analysis and belonged to MLST ST-16 (Figure 2, Appendix A).

### 2.3. Diversity and Phylogeny of C. botulinum Group I type B Strains and Their Botulinum Neurotoxin Genes

The eight sub-types of botulinum neurotoxin type B (B1 to B8) are less diverse than those of type A or type F, with an amino acid difference of 1.6–7.1%, and within sub-type differences from 0.1–2.1% [45]. The genome of 239 strains examined in the present study contained a type B neurotoxin gene (Figure 2); with 154 strains containing only a type B neurotoxin gene, and 85 strains containing an additional type A or type F neurotoxin gene (Figure 2). All type B neurotoxin genes were located in a *ha* neurotoxin gene cluster. Strains with genomes encoding botulinum neurotoxin type B belonged to the *C. botulinum* Group I lineage or the *C. sporogenes* lineage (Figure 2). Strains with a similar genomic background often possessed the same sub-type B neurotoxin gene but were also closely related to strains with other or no neurotoxin genes (Figure 2). The same sub-type B botulinum neurotoxin gene was also located in strains with distant genomic backgrounds, providing evidence for horizontal gene transfer.

The genome of 37 strains of *C. botulinum* Group I or *C. sporogenes* included a gene encoding botulinum neurotoxin sub-type B1 (Figure 2), and it was the only botulinum neurotoxin gene in these strains. Core genome SNP analysis revealed that these strains formed four clusters (Figure 2), that did not entirely align with the four sub-type B1 neurotoxin variants (Figure 4, Appendix A). The upper *C. botulinum* Group I cluster in Figure 2 includes closely related strains that belonged to MLST ST-30 (Appendix A) and were associated with foodborne or infant botulism in the USA (Appendix A). These strains possessed a gene encoding the same variant of sub-type B1 (second variant in Figure 4, Appendix A). Strains in the second *C. botulinum* Group I cluster belonged to MLST ST-34 (Figure 2, Appendix A), and were associated with foodborne botulism outbreaks in USA and UK non-clinical isolates. These strains possessed a gene encoding one of two sub-type B1 variants (either upper variant or lower variant (Figure 4)). Strains located within the *C. sporogenes* lineage exclusively possessed a gene encoding the third sub-type B1 variant (Figure 4, Appendix A), and separated into two clusters, both distantly related to sub-type B1 strains within the *C. botulinum* Group I lineage (Figure 2).

Core genome SNP analysis revealed that the 119 strains that possessed a sub-type B2 neurotoxin gene fell into eleven clusters that were very widely distributed amongst both the *C. botulinum* Group I and *C. sporogenes* lineages, and that these strains were often closely related to strains with another botulinum neurotoxin gene or no botulinum neurotoxin gene (Figure 2). The validity of these clusters was supported by MLST analysis (Appendix A). The presence of a sub-type B2 neurotoxin gene in many different genomic backgrounds indicates significant horizontal transfer of this botulinum neurotoxin gene between distantly related strains of both *C. botulinum* Group I and *C. sporogenes*. Genes encoding sub-type B2 neurotoxin have been identified in isolates from around the world, including strains associated with foodborne, infant and wound botulism (Figure 2). Some of these strains contained additional neurotoxin-encoding genes (Figure 2). Most strains possessed a gene encoding a complete sub-type B2 neurotoxin, although some strains possessed a gene encoding a truncated sub-type B2 neurotoxin. Strains with genes encoding a complete sub-type B2 neurotoxin (lower variant grouping in Figure 4, Appendix A) formed six clusters in *C. botulinum* Group I and two clusters in *C. sporogenes* (Figure 2). Fourteen isolates were from foodborne botulism outbreaks, including a 1998 UK outbreak associated with the consumption of home-preserved bottled mushrooms in oil brought back from Italy [66], a 2012 UK outbreak associated with consumption of olives from Italy (Table 1), and 2013 French outbreak associated with the consumption of home-canned beans [29]. Eighteen isolates were associated with infant botulism including three cases reported in the UK (Table 1) and one case in Greece [65]. Two further isolates were from wound botulism cases in Italy [59]. Non-clinical isolates included two strains isolated in our laboratory from Ugandan soil samples (Stringer SC and Peck MW, unpublished results), and other non-clinical isolates from UK, Poland, Spain, and USA (Appendix A). Some strains possessed genes encoding a truncated sub-type B2 neurotoxin and a complete sub-type A5 neurotoxin co-located in a *ha* neurotoxin gene cluster (Figure 2). Closely related variants of the truncated sub-type B2 neurotoxin were identified (upper grouping in Figure 4). Core genome SNP analysis established that these strains formed three clusters, with most strains closely related and belonging to MLST ST-47, including those associated with UK cases of wound botulism between 2004 and 2013 (Table 1), and a single strain associated with wound botulism in Switzerland in 2000 (Figure 2, Appendix A). Two additional strains (including one associated with foodborne botulism in France in 2011 [29]) were more distantly related according to core genome SNP analysis (Figure 2) and belonged to different MLST sequence types (ST-46 and ST-86).

Thirty-nine strains that possessed an entire or truncated sub-type B3 neurotoxin gene fell into three clusters within the *C. botulinum* Group I lineage, but not the *C. sporogenes* lineage (Figure 2). Most of these strains were associated with wound botulism (Figure 2). The upper cluster (Figure 2) contained A5B3 strains, with the gene encoding a truncated variant of the sub-type B3 neurotoxin (third variant in Figure 4, Appendix A) co-located with a sub-type A5 neurotoxin gene in an *ha* cluster. These strains were associated with UK outbreaks of wound botulism in 2000, 2004, 2006, and 2012 (Table 1). The central cluster in Figure 2 contained 26 strains with a gene encoding an entire sub-type B3 neurotoxin belonging to one of the upper two variants in Figure 4 (Appendix A). These strains were associated with outbreaks of wound botulism in UK (2006, 2009, 2010, 2014, 2015, and 2016; Table 1) and in Germany (2016). The 2016 wound botulism outbreaks in Germany and the UK will be described elsewhere in further detail [87]. The lower cluster in Figure 2 contained four strains with a gene encoding an entire sub-type B3 neurotoxin (lower variant in Figure 4, Appendix A), and one A2B3 strain with genes encoding an entire sub-type B3 neurotoxin (fourth variant in Figure 4, Appendix A) and neurotoxin sub-type A2. The B3 strains were associated with wound botulism in UK in 2009 (Table 1), and the A2B3 strain with foodborne botulism in Italy [59].

The 31 strains with an entire B5 neurotoxin gene were separated into three clusters within the *C. botulinum* Group I lineage, but not the *C. sporogenes* lineage (Figure 2). All but two of these strains contained an additional botulinum neurotoxin gene. The upper and centre clusters revealed by SNP analysis (Figure 2) contain A1B5 strains. These strains contained genes potentially encoding the sub-type B5 neurotoxin (upper sub-type B5 variant in Figure 4) and a sub-type A1 neurotoxin. Strains were associated with foodborne botulism in Ireland (2006) and the USA, infant botulism in the UK (1978) and USA, and wound botulism in UK (2009) [65]. In some A1B5 strains the encoded sub-type B5 neurotoxin is formed in small quantities, but biologically active sub-type B5 neurotoxin is not formed by a majority of strains [8,46,73,79]. These strains are referred to as A1b5 or A1(B5), respectively. The lower cluster revealed by SNP analysis (Figure 2) contained 11 strains that possessed a gene encoding one of two variants of sub-type B5 neurotoxin (centre and lower variants in Figure 4, Appendix A). This included B5, A2B5, A4B5 and B5F2 strains. Most of these strains were involved in infant botulism cases in UK (in 2012 and 2013; Table 1), Ireland (2014), Sweden and USA [8,25,46,66,82], but one B5F2 strain was associated with the largest UK outbreak of foodborne botulism (in 1989) involving the consumption of commercially-prepared hazelnut yoghurt (Table 1) [66,88].

The three strains that possessed a gene encoding botulinum neurotoxin sub-type B6 were located in a single cluster within the *C. sporogenes* lineage (see below for further details), with no strains belonging to the *C. botulinum* Group I (Figure 2). Previously, Smith et al. reported on eight sub-type B6 strains, with five strains found in the *C. sporogenes* lineage and three strains in the *C. botulinum* Group I lineage [23]. The genome of seven strains contained a gene encoding botulinum neurotoxin sub-type B7. Core genome SNP analysis assigned these strains to two clusters within *C. botulinum* Group I (Figure 2). The upper cluster contained six sub-type B7 strains and all belonged to MLST ST-34. Six (different) sub-type B7 strains associated with infant botulism in the eastern USA also belonged to MLST ST-34 [89]. The lower cluster contained a single A2B7 strain that had been associated with foodborne botulism in Italy [59]. Variants were identified of botulinum neurotoxin sub-types B6 and B7 (Figure 4).

### 2.4. Diversity and Phylogeny of C. botulinum Group I Type F Strains and Their Botulinum Neurotoxin Genes

Botulinum neurotoxin type F is highly diverse, with eight sub-types (F1 to F8) that differ by up to 36.2% of amino acid residues. Within sub-type differences are from 0.1–1.7% of amino acid residues [45]. Strains forming type F neurotoxin are less frequently associated with botulism than strains forming type A or B neurotoxin, and strains examined in the present study were associated with foodborne or infant botulism, but not wound botulism (Figure 2). A gene encoding botulinum neurotoxin type F was located in the genome of 33 strains within the *C. botulinum* Group I lineage (but none within the *C. sporogenes* lineage), including 17 strains with only a type F neurotoxin gene, and 16 strains with additional botulinum neurotoxin gene or genes (Figure 2). All genes encoding botulinum neurotoxin type F were located in an *orfX* neurotoxin gene cluster. Core SNP analysis revealed that all strains possessing a gene encoding neurotoxin sub-type F1 clustered together (Figure 2) and belonged to MLST ST-88 (Appendix A). All sequences of botulinum neurotoxin sub-type F1 were identical (Figure 5), including isolates derived from the first foodborne botulism outbreak associated with *C. botulinum* Group I type F that occurred in Denmark [90], and environmental (non-clinical) isolates. Strains possessing a gene encoding neurotoxin sub-type F2 clustered together (Figure 2) and belonged to MLST ST-7 and ST-14 (Appendix A). Genomes of these strains contained genes encoding botulinum neurotoxin sub-types F2 and B5, with two closely related variants of botulinum neurotoxin sub-type F2 identified (Figure 5, Appendix A). These strains were associated with foodborne botulism in UK (1989, Hazelnut yoghurt), and infant botulism in UK (2013), Sweden and USA [8,25,66,88].

All strains of *C. botulinum* Group I with genomes encoding botulinum neurotoxin sub-types F4 and/or F5 were non-clinical isolates from Argentina [25,56]. These strains formed five clusters (two each for F4 and F5 strains, and one including both F4 and F5 strains), indicating spread of these neurotoxin genes amongst bacteria with a diverse genomic background (Figure 2). The three clusters for strains with genes encoding botulinum neurotoxin sub-type F4 comprised: A2F4 (one strain); A2F4 and A2F4F5 (eight strains); and F4 (one strain). All the sequences of botulinum neurotoxin sub-type F4 were identical (Figure 5). The three clusters for strains with genes encoding botulinum neurotoxin sub-type F5 comprised: A2F4F5 (six strains); A2F5 (one strain); and F5 (one strain). All the sequences of botulinum neurotoxin sub-type F5 were identical (Figure 5, Appendix A).

### 2.5. Diversity and Phylogeny of Strains within the C. sporogenes Lineage and Their Botulinum Neurotoxin Genes

One hundred and four isolates were assigned to the *C. sporogenes* lineage according to core genome SNP analysis (Figure 2). This included several strains previously designated as *C. botulinum* Group I, on the basis of their ability to form botulinum neurotoxin. Whilst the genome of a majority of isolates assigned to the *C. sporogenes* lineage lacked a botulinum neurotoxin gene, the genome of twenty isolates possessed a gene that encoded botulinum neurotoxin of either sub-type B1, B2 or B6. These genes have previously been reported in strains of *C. sporogenes* [8,22,23,24,25]. Examples of strains of *C. sporogenes* possessing a botulinum neurotoxin gene that were associated with foodborne, infant and wound botulism are given in Table 2, whilst other strains possessing a botulinum neurotoxin gene had no known association with botulism. Strains of *C. sporogenes* with a gene encoding botulinum neurotoxin type B were isolated over a wide geographical area (Appendix A).

All sub-type B1, B2 and B6 botulinum neurotoxin genes were located in a *ha* neurotoxin gene cluster (Figure 2). The variant of sub-type B1 neurotoxin encoded by *C. sporogenes* (third variant in Figure 4, Appendix A) can be distinguished from the three variants of sub-type B1 neurotoxin encoded in the genome of strains of *C. botulinum* Group I (Figure 4). The variant of sub-type B2 neurotoxin encoded by most *C. sporogenes* strains is identical to that encoded by many strains of *C. botulinum* Group I (Figure 4). However, one variant is uniquely encoded by *C. sporogenes* strain B2 450 (CB1071), and an unusual divergent variant is encoded only by *C. sporogenes* strain ATCC 51387 (CB1056) (Figure 4). In the present study, genes encoding botulinum neurotoxin sub-type B6 were located in *C. sporogenes*, but not *C. botulinum* Group I (Figure 2). Genetic analysis carried out previously and also in the present study showed that the botulinum neurotoxin genes encoded by *C. sporogenes* are primarily carried on a plasmid [8,22,24,25,44,91,92]. These strains may lose their ability to form botulinum neurotoxin, and this phenomenon has been examined further in the present study with strain 2345, where the genome sequences of the original toxic strain (CB0594) and a non-toxic isolate derived in our laboratory (CB0595) have been compared. In the original toxic strain (CB0594), the neurotoxin gene is located on a plasmid very homologous to pCBH of strain Prevot 594 (https://www.ncbi.nlm.nih.gov/nucleotide/CP006901.1). This is a large plasmid (257 kb), and CB0594 has eight contigs mapping onto that plasmid, for a total of 240 kb (including the 15 kb contig with the neurotoxin gene cluster). All of these are missing in CB0595, indicating that the inability to form neurotoxin is due to plasmid loss. An important, but unresolved question, is why do most *C. botulinum* Group I strains possess a gene encoding botulinum neurotoxin, but only few C. sporogenes strains? The answer may be low neurotoxin gene stability (as noted for strain 2345). A further important question is whether a non-toxigenic strain of *C. sporogenes* (or *C. botulinum* Group I) may acquire a type B neurotoxin gene through horizontal gene transfer.

The *C. sporogenes* lineage separates into three clusters, each containing isolates that possess and lack a gene encoding a botulinum neurotoxin (Figure 2). Some of these isolates have been previously reported to belong to the *C. sporogenes* lineage [23,25]. Isolates in the upper *C. sporogenes* cluster, that possessed a gene encoding botulinum neurotoxin sub-type B2, had been associated with foodborne botulism involving ham in France (Table 2, Figure 2). Interestingly, the sub-type B2 strain F11 that was associated with animal (cattle) botulism in France also belonged to MLST ST-54 [29], as did nine sub-type B2 strains and two non-toxic strains that are located in the upper *C. sporogenes* cluster in the present study (Figure 2). The central *C. sporogenes* cluster included isolates that possessed a gene encoding botulinum neurotoxin sub-type B1, B2 or B6, and had been involved in foodborne botulism (Australia), infant botulism (Australia and USA), and wound botulism (Italy) (Table 2, Figure 2). Two strains (Osaka05 and Okayama2011) not included in the present study, that were associated with infant botulism in Japan and formed sub-type B6 neurotoxin, were distantly related to strains of *C. botulinum* Group I and also belong to the *C. sporogenes* lineage [7,24,25,46,92]. Interestingly, the lower *C. sporogenes* cluster has not been described previously and contains five isolates with a gene encoding botulinum neurotoxin sub-type B1. It is more distantly separated from the first two clusters (Figure 2). Strains R1125/03 and R1135/03 were associated with a fatal outbreak of foodborne botulism in the UK in 2003 involving the consumption of a home-prepared meat dish (“bigos”) brought from Poland (Table 1 and Table 2, Figure 2). Two Polish men shared this meal; one developed botulism in the UK and subsequently died, while the second was diagnosed with botulism in Poland [66]. Also present in the lower cluster is strain 2113/01, a non-clinical UK isolate from an unopened can of infant formula that possessed a gene encoding botulinum neurotoxin sub-type B1 [93,94]. Jacobson et al. [68] reported on four additional strains located in MLST STs associated with the *C. sporogenes* lineage. Three strains (14842, 14843 and 14844) belonged to ST-5, and one strain (A207) belonged to ST-17. Those belonging to ST-5 were reported to be A(B) strains with a unique neurotoxin cluster arrangement, while the ST-17 strain was reported to form sub-type A1 neurotoxin [68]. It was noted that these strains were distinct from the other *C. botulinum* Group I strains analysed [68]. The present study and previous work [24] both place ST-5 and ST-17 within the *C. sporogenes* lineage.

### 2.6. Impact of Genomic and Physiological Variability on the Botulism Risk

Bacterial genomics, the identification of distinct lineages and clusters, and the characterisation of strains within distinct lineages/clusters can make an important contribution to improved risk assessments and the prevention of botulism. Several clusters identified in the present study demonstrate a bias in terms of association with foodborne, infant or wound botulism, or are dominated by non-clinical (environmental) isolates (Figure 2). For example, many strains in the upper cluster of sub-type A2 strains are associated with infant botulism, while all strains in the lower cluster of sub-type A2 strains are non-clinical (environmental) isolates (Figure 2). Strains associated with foodborne botulism were widely distributed amongst many clusters, while strains associated with wound botulism were located in a relatively limited number of clusters (Figure 2). From this it might be inferred that strains possessing botulinum neurotoxin-encoding genes in certain clusters present a greater risk of botulism, or even certain types of botulism, than strains in other clusters. However, it could also be the case that the patterns observed merely represent opportunity, and that all strains that form botulinum neurotoxin have a similar potential to cause foodborne, infant or wound botulism. Investigating this further is not part of the present study but represents an important area for future endeavours that could include genome-wide association studies allied with physiological studies to dissect the mechanisms key to pathogenicity and pathogen transmission.

There are, however, two pieces of evidence that suggest that strains within the *C. botulinum* Group I lineage currently present a greater botulism risk than strains within the *C. sporogenes* lineage. Firstly, a greater fraction of strains within the *C. botulinum* Group I lineage have been associated with foodborne, infant and wound botulism, and secondly a greater proportion of strains in the *C. botulinum* Group I lineage possess a gene encoding botulinum neurotoxin (Figure 2). However, detailed information on physiological differences between neurotoxin-forming strains in the *C. botulinum* Group I and *C. sporogenes* lineages is presently lacking, and it is conceivable that circumstances may arise in the future (e.g., new approaches to food processing) that favour strains in the *C. sporogenes* lineage. A detailed analysis of the physiological differences between neurotoxin-forming strains in the *C. botulinum* Group I and *C. sporogenes* lineages is needed to establish whether it is necessary to conduct separate risk assessments for neurotoxigenic strains of *C. botulinum* Group I and *C. sporogenes*.

*C. botulinum* Group I spores are of high thermal resistance and the target of the Botulinum cook (121 °C/3 min) given to low acid canned foods. For many decades, *C. sporogenes* has been used as a surrogate for *C. botulinum* Group I in food sterilisation tests [7,14,17,19]. In particular, spores of the nontoxigenic putrefactive anaerobe strain PA3679 are reported to have a higher thermal resistance than spores of *C. botulinum* Group I and are widely used in the validation of thermal processes in the food industry. This strain is generally referred to as a strain of *C. sporogenes* and does not possess a botulinum neurotoxin gene. There appear to be several versions of strain PA3679. Weigand et al. reported their version to align with *C. sporogenes* strains, while Brown et al. reported that their version was more closely related to strains of *C. botulinum* Group I than to strains of *C. sporogenes* [14,24]. Further studies carried out on a number of isolates labelled “PA3679” confirmed strain heterogeneity. While some isolates (e.g., 1961-62, 2007) had low spore thermal resistance and genetically resembled *C. sporogenes*, other isolates (e.g., Camp, UW, FDA, 1961-64) had high spore thermal resistance and genetically resembled *C. botulinum* Group I [7,19]. In the present study, strain PA3679 isolate Camp (CB1254) was located in the *C. botulinum* Group I lineage (Figure 2); thus, this isolate appears to be a non-toxigenic strain of *C. botulinum* Group I rather than a strain of *C. sporogenes*.

*Clostridium* spores include a large store of pyridine-2, 6-dicarboxylic acid (DPA) in a 1:1 chelate with calcium ions within their core which promotes spore thermal resistance [95]. DPA uptake into the spore core is regulated by the *spoVA* operon [96,97]. Recently a *spoVA2mob* operon on a transposon identified in *Bacillus* was associated with a raised DPA content and an increase in spore thermal resistance [98]. Strain PA3679 isolate Camp had a second *spoVA* operon, and this was identified as one factor contributing to the elevated thermal resistance of spores of this strain [12]. In the present study, bioinformatic analysis revealed that a second *spoVA* operon (*spoVA2*) was present in 56% of strains in the *C. botulinum* Group I lineage, but only 3% of strains in the *C. sporogenes* lineage (Appendix A, Appendix A), potentially alluding to a high spore thermal resistance for many strains in the *C. botulinum* Group I lineage. Strains with a second *spoVA* operon (*spoVA2*) formed five clusters in the *C. botulinum* Group I lineage and one cluster in the *C. sporogenes* lineage (Appendix A, Appendix A). When conducting challenge tests or validating thermal processes for *C. botulinum*, it is therefore prudent to select appropriate strains from *C. botulinum* Group I and/or *C. sporogenes* that takes account of genomic/physiological variability. Toxic and non-toxic strains have been identified in both lineages. The use of strains located within the *C. sporogenes* lineage may not be appropriate for thermal death studies.

### 2.7. Pan-genome Comparison of C. botulinum Group I and C. sporogenes Genomic Clusters and Identification of C. botulinum Group I- and C. sporogenes-specific Genes

The 556 genomes examined in this study separated into a distinct *C. botulinum* Group I lineage, consisting of 452 genomes, and a *C. sporogenes* lineage that comprised two previously known clusters of 94 genomes and an additional new cluster of ten genomes (Figure 2). In a previous study, Williamson et al. [56] provided primers to distinguish *C. botulinum* Group I from *C. sporogenes*. When tested in silico against all genomes used in the present study, the primer pair correctly distinguished the previously known *C. botulinum* Group I and *C. sporogenes* clusters, but failed to correctly identify the additional (lower) *C. sporogenes* cluster, as strains within this cluster were positive for the *C. botulinum* PCR product (10/10) and negative for the *C. sporogenes* PCR product (9/10). Similarly, Weigand et al. [24] performed genome analysis for *C. botulinum* Group I and C. *sporogenes*, and identified 11 candidate genes for a species-specific PCR. These were also tested against the 452 *C. botulinum* Group I and 104 *C. sporogenes* genomes used in the present study. None of the primer sets was able to correctly amplify all genomes tested; the *nifS* primers detected all *C. sporogenes* genomes, but also amplified 29 of the *C. botulinum* Group I genomes, whereas the *spoVB* primers correctly identified 452 *C. botulinum* Group I and 94 *C. sporogenes* genomes, but failed to amplify the additional (lower) *C. sporogenes* cluster. As both genes were present in all genomes, it might be that the efficiency could be improved by the selection of new primers. The other nine genes [24], performed less well, either amplifying a larger number of *C. botulinum* genomes (*thiHG*), or not or only partially amplifying the new *C. sporogenes* cluster.

Pangenome analysis was used to search for genes for *C. botulinum* Group I and *C. sporogenes*. The pangenome of the combined 556 genomes was generated using Prokka-annotated genome sequences and the Roary software package [99], followed by identification of lineage-specific genes defined. The pangenome consisted of 18,731 annotated features, with 2420 annotated features present in >95% of the genomes (core genome). The distribution of genes linked to the position in the core genome-based phylogenetic tree is shown in Figure 6A. Due to the different sizes of the groups, lineage specificity was initially set at present in ≥90% (over-represented) or present in ≤10% (under-represented) for *C. botulinum* Group I, and ≥80%/≤20% for the larger 94-genome *C. sporogenes* cluster, and ≥70%/≤30% for the smaller 10-genome *C. sporogenes* cluster. This identified 224 features over-represented in *C. botulinum*, and 79 features over-represented in *C. sporogenes*. The relative position of features specific for *C. botulinum* Group I and *C. sporogenes* were plotted using the complete genomes of *C. botulinum* Group I strain ATCC 19397 and *C. sporogenes* strain NCIMB 10696 (Figure 6B), and these features were spread throughout the chromosome, similar to the two *C. botulinum* Group II lineages [2].

Of the over-represented features for *C. botulinum* Group I and *C. sporogenes*, 35 showed a perfect distribution for *C. botulinum* and 14 did likewise for *C. sporogenes*. These were further tested at the nucleotide level using BLAST, resulting in 13 genes specific for *C. botulinum* Group I and four genes specific for *C. sporogenes*, defined by >99% specificity (Appendix A). These genes comprise candidates for a future PCR assay to distinguish *C. botulinum* Group I and *C. sporogenes*, but the development and testing of such an assay falls outside the scope of this study.

## 3. Conclusions

This comparative genomic investigation of 556 highly diverse strains of *C. botulinum* Group I and *C. sporogenes* (including 417 newly sequenced strains) has analysed population diversity, structure, and spread, and established novel relationships between whole genome lineage, botulinum neurotoxin sub-type variant, association with foodborne, infant and wound botulism, epidemiology, and geographic origin. The impact of genomic and physiological variability on the botulism risk has been evaluated. Strains had up to three botulinum neurotoxin genes (of types A, B and/or F), but no novel botulinum neurotoxin genes (serotypes or sub-types) were discovered. This may suggest a limited number of unknown botulinum neurotoxin genes (serotypes or sub-types) within the genomes of *C. botulinum* Group I and *C. sporogenes*. The present study described novel botulinum neurotoxin sub-type variants, each with a unique amino acid sequence. Core genome SNPs analysis identified two major lineages; *C. botulinum* Group I (most strains possessed neurotoxin gene(s) of types A, B and/or F) and *C. sporogenes* (some strains possessed a type B neurotoxin gene). Both lineages contained strains responsible for foodborne, infant and wound botulism. An additional new *C. sporogenes* cluster was discovered that included five strains with a gene encoding botulinum neurotoxin sub-type B1. There was significant evidence of horizontal transfer of botulinum neurotoxin genes between distantly related bacteria. Each lineage also included strains that lacked a botulinum neurotoxin-encoding gene. The pan-genome of *C. botulinum* Group I and *C. sporogenes* contained 18,731 annotated features, with a core genome of 2420 annotated features (present in >95% of genomes), and 224 and 79 features over-represented in the genomes of *C. botulinum* Group I and *C. sporogenes*, respectively. There were no integration hotspots, with lineage-specific features located throughout the genomes. The genome sequences are also a valuable resource to future research endeavours, for example, on pathogen biology, the evolution of bacteria and their neurotoxin genes, and improved pathogen detection and discrimination. When investigating outbreaks of human or animal botulism or carrying out surveillance studies, it is vital to establish the *C. botulinum* Group and lineage of the bacterium involved, and the associated botulinum neurotoxin(s). The increased availability of whole genome sequences for *C. botulinum* Group I and *C. sporogenes* (this study) and *C. botulinum* Group II (previously [2]) gives a greater value to genetic tests. Improved PCR tests can be used to rapidly detect and identify the bacterium present, its lineage, and botulinum neurotoxin gene(s). While the genome sequences enable enhanced isolate discrimination and detailed epidemiological investigation. This may support enhanced risk assessments and prevent and/or minimise botulism outbreaks.

## 4. Materials and Methods

### 4.1. Strains of C. botulinum Group I and C. sporogenes

The genomes of 556 strains were examined (417 newly sequenced strains and 139 genomes from public sources, Appendix A). The newly sequenced strains were isolated in the authors’ laboratory or kindly provided by colleagues (B. Dorner, T. Grenda, C. Hatheway, V. Broussolle, A. East, P. McClure, T. Roberts, H. Tranter, J. Crowther, P. Barrett, L. Taylor and M. Wictome; Appendix A). The strains were isolated over a period of more than 100 years, from 24 countries throughout Europe, Asia, Africa, Australasia, north America and south America (Figure 1). Seventy-three strains were isolated following outbreaks of foodborne botulism, 70 strains were associated with infant botulism, 82 strains were associated with wound botulism, and 139 strains were non-clinical isolates not associated with human botulism (Appendix A). The status of the remaining strains is unknown. Based on their genome sequence, 452 strains were assigned to *C. botulinum* Group I and 104 strains to *C. sporogenes*. This led to some strains now being considered as *C. botulinum* Group I and not *C. sporogenes*, or vice-versa. Most of the strains assigned to *C. botulinum* Group I possessed a botulinum neurotoxin gene or genes, while some of the strains assigned to *C. sporogenes* possessed a botulinum neurotoxin gene. The strains possessed up to three botulinum neurotoxin genes (Table 3). In this study, strains are referred to by their type or sub-type of botulinum neurotoxin gene(s). Thus, for example, B5F2 strains possess genes encoding botulinum neurotoxin sub-types B5 and F2. It was not established whether all neurotoxin genes were associated with formation of botulinum neurotoxin.

### 4.2. Genomic DNA Preparation and Whole Genome Sequencing

DNA extraction was performed by following the traditional Gram-positive cell lysis procedures [2]. Whole genome sequencing of the *C. botulinum* gDNA samples was carried out at the Earlham Institute, UK [2]. Briefly, libraries were assembled with 1ng of DNA. Libraries were constructed using unique 9 bp dual index combinations allowing samples to be multiplexed. Libraries were pooled then sequenced with a 2 × 250 bp read metric on an Illumina HiSeq 2500 sequencer.

### 4.3. Genome Assembly and Quality Control

Shovill version 1.0.4 (https://github.com/tseemann/shovill) with the Spades 3.13.0 assembler module [100] were used to assemble genome sequences, with contigs smaller than 200 bp and coverage lower than five-fold excluded. Assembly metrics were obtained using QUAST version 4.6 [101] with genome size, N50, L50 and number of contigs used for quality assessment of the assemblies. Genome assemblies were annotated using Prokka version 1.13 [102]. Multilocus Sequence Typing (MLST) was done using mlst version 2.16.0 (https://github.com/tseemann/mlst) and the sequence profiles from the *C. botulinum* PubMLST homepage (https://pubmlst.org/cbotulinum/). The Genbank/SRA accession numbers for genome assemblies, FASTQ reads, and source information are provided in Appendix A.

### 4.4. Identification of Botulinum Neurotoxin Sub-Types and Accessory Protein Configuration

The Prokka annotated features were searched with representatives of the A, B, C, D, E, F, G, FA, X and Ebont/J toxins [45,69] as described in Brunt et al. [2]. Gene sequences representing incomplete, fragmented and inactivated toxin genes were manually extracted from the individual genome sequences, and frameshifts/stop codons manually repaired to obtain the amino acid sequences for comparison. Toxins were assigned to specific sub-types by alignment of the amino acid sequences with the toxin reference types [45] using the MUSCLE module of MEGA7 [72], and subsequent generation of a Neighbour-Joining tree and clustering with reference toxins, with Gaps/Missing data set to pairwise deletion. Accessory genes were identified using BLAST using reference sequences [3].

### 4.5. Pangenome Analyses, Target Gene Identification and in silico PCR

The pangenome of *C. botulinum* Group I and closely related *C. sporogenes* was identified using Roary version 3.13 [99] and was based on the Prokka-annotated genome assemblies, with 80% and 90% BLAST cut-off percentages, and paralog clustering both switched off and on [103], as described previously for *C. botulinum* Group II [2]. Briefly, the Roary-outputs were analysed using Scoary [104] with a Bonferroni corrected *p*-value cut-off of 0.05. For *C. botulinum* Group I, genes were considered over- or under-represented if either present in ≥90% of the tester group and less than ≤10% of the comparator group, or vice versa. However, due to the different group size, lineage specificity was initially set at ≥80%/≤20% for the larger 94-genome *C. sporogenes* cluster, and ≥70%/≤30% for the smaller 10-genome *C. sporogenes* cluster. The plots for the pangenome distribution were generated with roary plots (https://github.com/sanger-pathogens/Roary/tree/master/contrib/roary_plots). *In silico* PCR was performed using MIST [105].

### 4.6. Core Genome SNPs Analysis and Phylogenetic Trees

Phylogenetic trees were generated using core genome SNPs and the ParSNP program [70] with settings described previously [106]. Phylogenetic trees were annotated using Figtree (http://tree.bio.ed.ac.uk/software/figtree/) and MEGA7 [72].

## Figures and Tables

**Figure 1 toxins-12-00586-f001:**
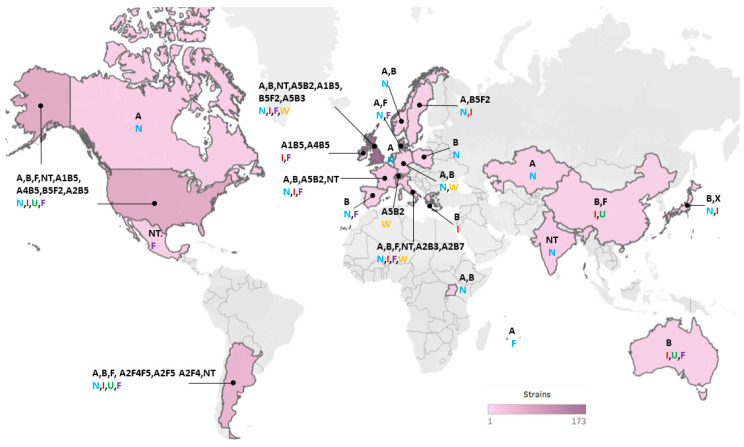
Geographical locations and heat map of *C. botulinum* Group I and *C. sporogenes* strains included in the present study. A total of 359 strains were attributed to a geographical location. Further details about individual isolates are given in Appendix A. For each specified country, the detected botulinum neurotoxin genes for isolates are indicated (in black; with NT = no toxin gene), and the types of botulism are shown in colour (N = non-clinical; I = infant botulism; U = unknown; F = foodborne botulism; W = wound botulism).

**Figure 3 toxins-12-00586-f003:**
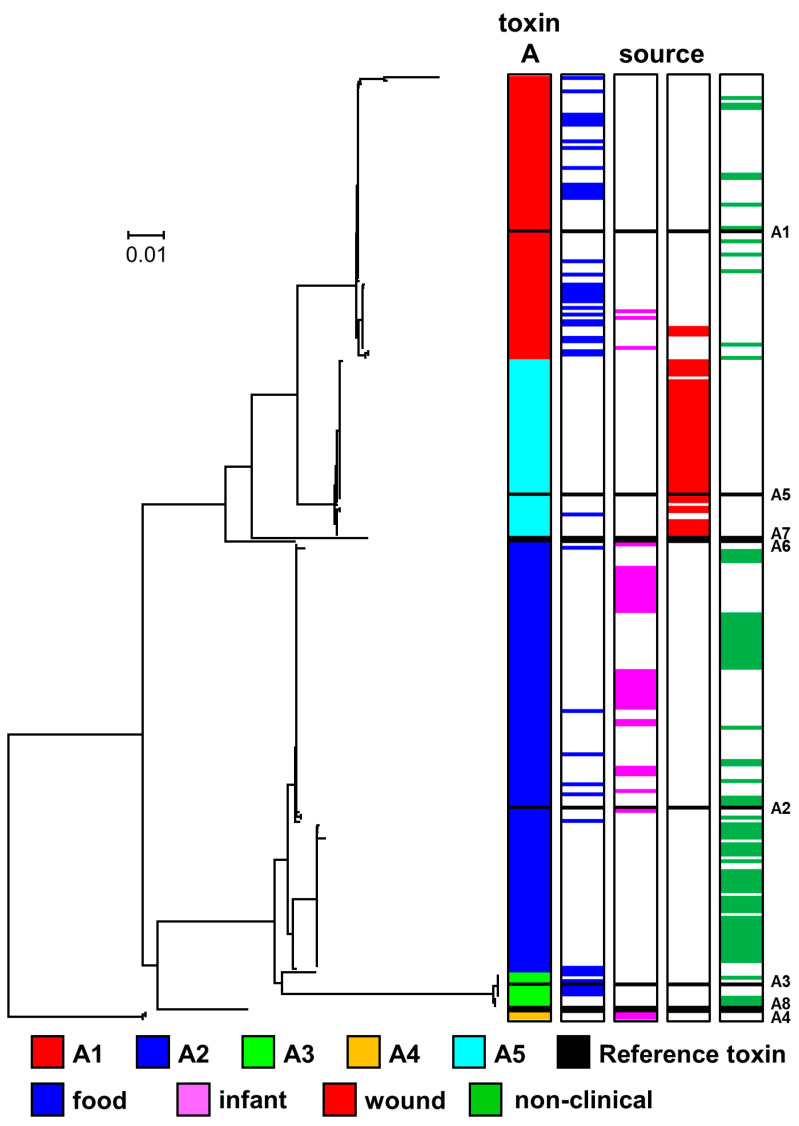
Phylogeny of sub-types of botulinum neurotoxin type A. The toxin protein sequences were aligned with the Muscle module of MEGA7 [72] algorithm, and the phylogenetic tree was generated using the Neighbour-Joining method. Variants of five type A neurotoxin sub-types were identified. Scale bar represents the number of amino acid substitutions per site. A total of 271 amino acid sequences were analysed. White blocks represent absence of information regarding source. Black blocks represent reference neurotoxins [45]. Further details about individual sequences and isolates are given in Appendix A, where they appear in the same order as in this Figure. The sources of the isolates are given in Appendix A.

**Figure 4 toxins-12-00586-f004:**
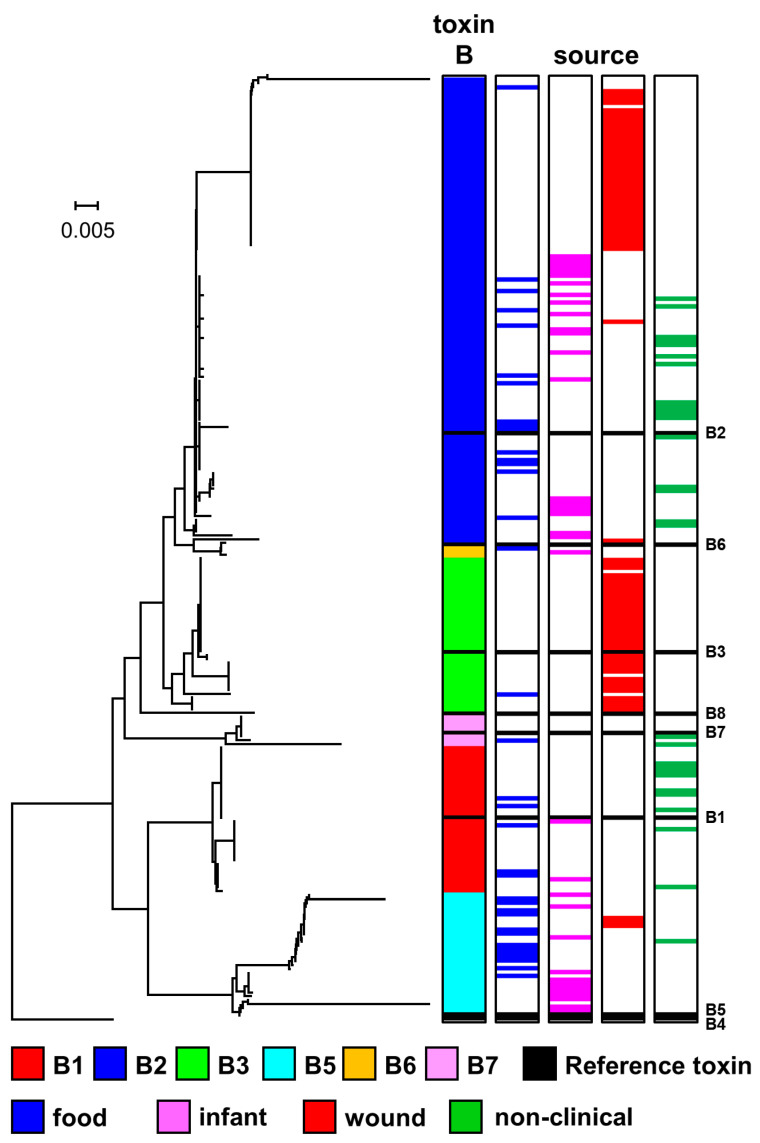
Phylogeny of sub-types of botulinum neurotoxin type B. The toxin protein sequences were aligned with the Muscle module of MEGA7 [72] algorithm, and the phylogenetic tree was generated using the Neighbour-Joining method. Variants of six type B neurotoxin sub-types were identified. Scale bar represents the number of amino acid substitutions per site. A total of 239 amino acid sequences were analysed. White blocks represent absence of information regarding source. Black blocks represent reference neurotoxins [45]. Further details about individual sequences and isolates are given in Appendix A, where they appear in the same order as in this Figure. The sources of the isolates are given in Appendix A.

**Figure 5 toxins-12-00586-f005:**
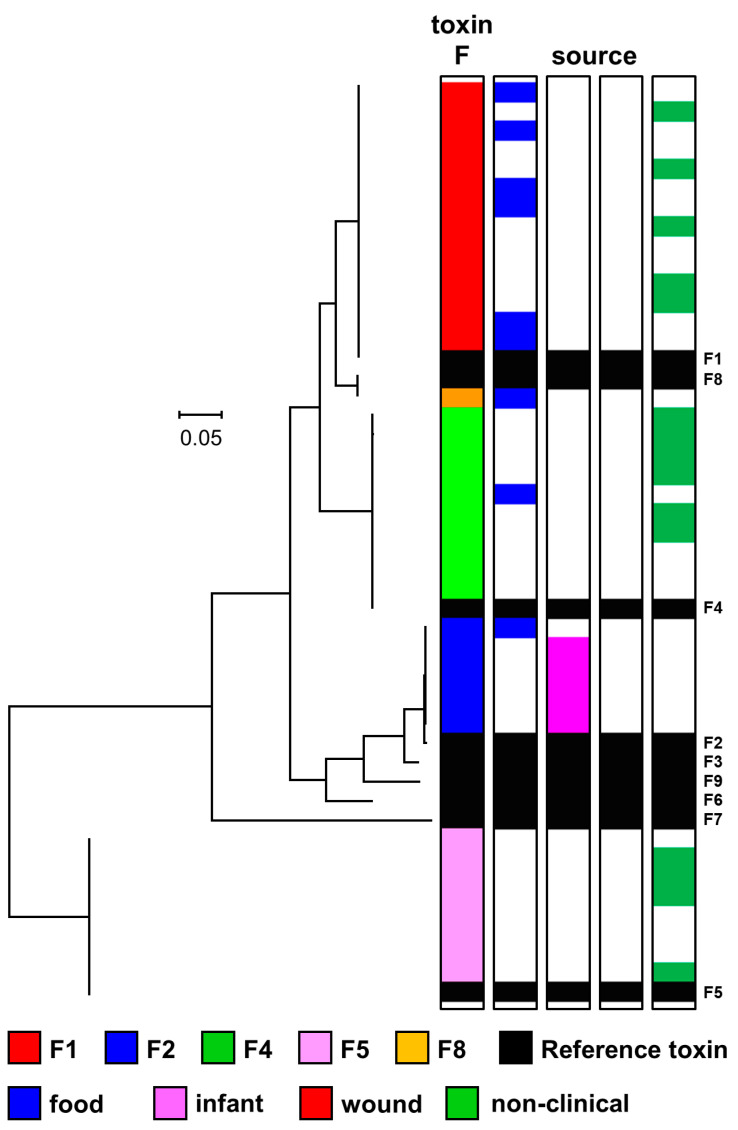
Phylogeny of sub-types of botulinum neurotoxin type F. The toxin protein sequences were aligned with the Muscle module of MEGA7 [72] algorithm, and the phylogenetic tree was generated using the Neighbour-Joining method. Variants of five type F neurotoxin sub-types were identified. Scale bar represents the number of amino acid substitutions per site. A total of 39 amino acid sequences were analysed. White blocks represent absence of information regarding source. Black blocks represent reference neurotoxins [45]. Further details about individual sequences and isolates are given in Appendix A, where they appear in the same order as in this Figure. The sources of the isolates are given in Appendix A.

**Figure 6 toxins-12-00586-f006:**
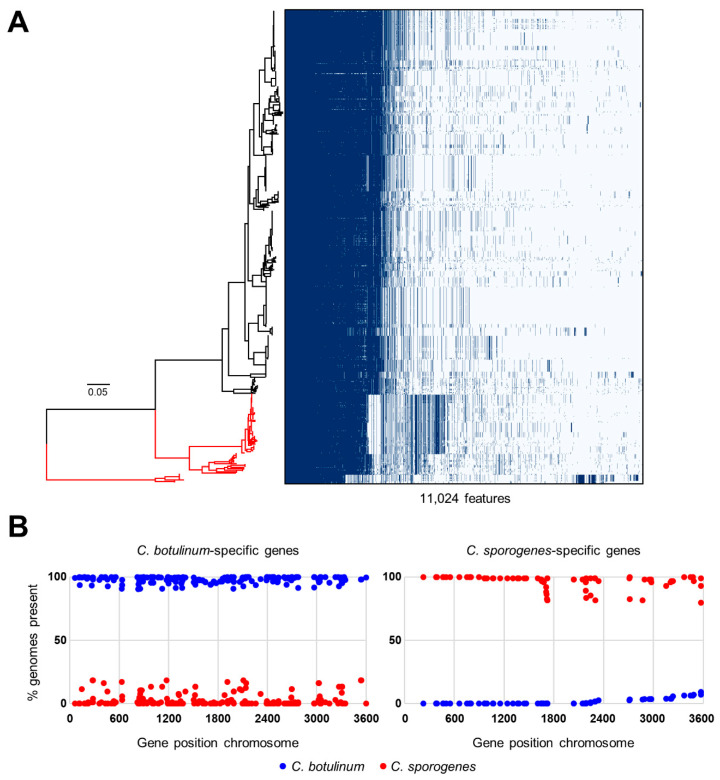
Pan-genome comparison of the *C. botulinum* Group I and *C. sporogenes* lineages (Figure 2). (**A**) Graphical representation of feature distribution linked to the SNP-based phylogenetic tree (Figure 2). (**B**) The lineage-specific genes are distributed throughout the genome, as shown using representative genomes for the *C. botulinum* Group I lineage (ATCC 19397) and the *C. sporogenes* lineages (NCIMB 10696). Dark blue circles represent *C. botulinum* Group I lineage features, and red circles represent *C. sporogenes* lineage features.

**Table 1 toxins-12-00586-t001:** Summary of botulism cases recorded in the UK between 1989 and 2019. ^a^ Confirmed case: diagnosis of botulism confirmed by laboratory testing (detection of botulinum toxin by bioassay or gene by PCR). Probable case: a clear clinical diagnosis of botulism, laboratory tests either not undertaken or negative (e.g., suboptimal or delayed sampling). ^b^ Early UK cases of foodborne botulism and infant botulism were summarized previously [65,66]. ^c^ unknown. ^d^ not tested in the present study. ^e^
*Clostridium botulinum* Group II sub-type B4 [2]. ^f^
*Clostridium butyricum*. ^g^ Wound botulism (injectable drug use) data include the number of confirmed cases, with number of probable cases in parenthesis. The number of confirmed cases for each botulinum toxin type from the outbreak investigation, and the number of sequenced isolates for each botulinum toxin gene sub-type are shown in parentheses.

Year	Confirmed Cases (Probable Cases) ^a^	Botulinum Toxin Type(s) From Outbreak Investigation	Botulinum Toxin Gene Sub-type(s) Detected in This Study	Comments
Foodborne botulism ^b^
1989	27	B	B5F2	Commercial hazelnut yoghurt
1998	2	B	B2	Bottled mushrooms (Italy)
2003	1	B	B1	Sausage (Poland)
2004	1	un ^c^	nt ^d^	Hummus
2004	1	A	nt	Recent travel to Georgia
2005	1	B ^e^	nt	Home preserved pork (Poland)
2010	1	B	nt	Recent travel to Algeria
2011	3	A	A1	Commercial korma sauce
2012	1	B	B2	Olives (Italy)
2013	1	un	nt	Mushroom (Poland)
2016	1	B	nt	Tuna (Italy)
Infant botulism ^b^
1989	1	B	nt	Travel to Yemen, fed honey
1993	1	B	B2	Travel to Spain
1994	1	A	nt	Fed honey, weaning
2001	1	B	nt	Weaning, infant formula milk
2007	1	A	A2	Weaning
2007	1	B	B2	Weaning
2009	1	A	A2	Fed honey
2009	1	A	A2	Fed honey
2010	1	E ^f^	nt	Fed honey, kept terrapins
2011	1	A	A2	Fed honey
2011	1	A	A2	Fed honey
2011	1	B	B2	Breast and bottle fed, dusty home
2012	1	B	B5	Travel, dusty hostel rooms
2013	1	Bf	B5F2	Fed honey
2013	1	Bf	B5F2	Weaning
2017	1	B	nt	Travel to Western USA
2018	1	B	nt	Weaning
Intestinal colonisation botulism
2018	1	un	nt	Bowel obstruction, several co-morbidities
Wound botulism ^g^
2000	3 (2)	A	A5B3(1)	
2001	3 (1)	A	nt	
2002	14 (6)	A(12), B(2)	nt	
2003	7 (8)	A(6), AB(1)	nt	
2004	15 (26)	A(13), B(1), AB(1)	A5B2(33), A5B3(1)	
2005	6 (22)	A(5), B(1)	A5B2(2)	
2006	9 (21)	A(6), B(1), AB(2)	A5B2(1), A5B3(4), B3(1)	
2007	3	A(3)	A5B2(2)	
2008	4	A(1)	nt	
2009	12 (8)	A(4), B(7), un(1)	A1B5(3), B3(5)	
2010	3	A(1), B(1), un(1)	A5B2(1), B3(2)	
2012	1 (2)	A	A5B3(1)	
2013	2 (2)	A (1), B (1)	A5B2(1)	
2014	2 (3)	B	B3(3)	
2015	19 (28)	B(16), un(2)	B3(13)	Large outbreak related to batch of contaminated heroin
2016	5 (2)	B	B3(1)	
2017	1 (1)	B	nt	
2018	2 (2)	B	nt	
2019	1 (1)	B	nt	

**Table 2 toxins-12-00586-t002:** Examples of strains within the *C. sporogenes* lineage associated with outbreaks of botulism.

Strain	*C. sporogenes* Cluster *	Botulinum Neurotoxin Gene	Type of Botulism	Country (Year)	Reference
Prevot 594	upper	B2	Foodborne (ham)	France (1951)	[46]
2345	upper	B2	Foodborne (ham)	France (1961)	[29]
CDC 1632	centre	B1	Infant	USA (1977)	[91]
B2 450	centre	B2	Wound (drug abuse)	Italy (2009)	[59]
AM370	centre	B6	Foodborne (salted fish)	Australia (1979)	[25]
AM1195	centre	B6	Infant	Australia (1987)	[25]
R1125/03	lower	B1	Foodborne (meat)	UK (2003)	[66]
R1135/03	lower	B1	Foodborne (meat)	UK (2003)	[66]

* According to Figure 2.

**Table 3 toxins-12-00586-t003:** Neurotoxin genes present in the genomes of isolates of *C. botulinum* Group I and *C. sporogenes* (see Appendix A for further details of the isolates).

Botulinum Neurotoxin Gene Sub-Type	Number of Strains Included in This Study
A1	64
A1 B5	20
A2	114
A2 B2	1
A2 B3	1
A2 B5	1
A2 B7	1
A2 F4	3
A2 F5	1
A2 F4 F5	6
A3	9
A4 B5	2
A5 B2	44
A5 B3	8
B1	37
B2	73
B2 FA	1
B3	30
B5	2
B5 F2	6
B6	3
B7	6
F1	14
F4	1
F5	1
F8	1
none	107
TOTAL	556

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
