# Peer review of "Diversity of the Genomes and Neurotoxins of Strains of Clostridium botulinum Group I and Clostridium sporogenes Associated with Foodborne, Infant and Wound Botulism"

_toxins, 2020, doi:10.3390/toxins12090586_

Round 1

Reviewer 1 Report

The reviewed article is the first such a comprehensive study on genetic features of C. botulinum Group I and C. sporogenes isolates from 24 countries and six continents. The mentioned strains were collected from epidemiological investigations associated with foodborne, infant and wound botulism cases. Strains were isolated over a period of above 100 years. The experiment on such a wide scale has not been conducted up to now. The study provided complete characteristic of genetic features of C. botulinum Group I and C. sporogenes strains. Authors thoroughly described botulinum neurotoxin sub-types in C. botulinum and C. sporogenes strains. Moreover, they discovered new neurotoxin sub-type variants in C. sporogenes strains. It is worthy to emphasise that authors evaluated the genomic impact (genome lineage and neurotoxin sub-types) on the risk of different types of botulism occurrence. In my opinion this study will significantly increase the availability of completely characterized whole genome sequences and will enhance to improve diagnostic tests elaboration and complex risk assessment in order to prevent botulism outbreaks. 

I have only one kind suggestion: p. 16, line 507 - please, change the name of "bigosh" into proper Polish spelling "bigos".

Author Response

Authors response: Thank you for the thorough review, positive feedback, and helpful suggestion.

I have only one kind suggestion: p. 16, line 507 - please, change the name of "bigosh" into proper Polish spelling "bigos".

Authors response: The requested correction has been made.

Reviewer 2 Report

The author(s) has newly sequenced 417 strains of C. botulinum group I and C. sporogenes, and performed a comparative genome study with total 556 diverse strains (417 newly sequenced strains and 139 previously sequenced strains).

This manuscript is well written and includes the important information on genetics of C. botulinum and C. sporogenes. I have just a few comments listed below.

  1. P1L26 Names of bacteria should be in italics.

  1. P2L77 Please add a definition of “B5F2”, “A5B3”…, notations of toxin types in two large characters.

  1. P3L102 C. botulinum strain ATCC3502 is also known as strain Hall. It would be helpful to include this information.

  1. P3L117 Fig.1 Please add explanation of abbreviations, “N”, “I”, “F”, “W”, “U” in legend.

  1. P6 Fig.2 Please add explanation of the right side black column “new” in legend.

  1. P9L261 Please include enough information about “variant group” of MLST.

  1. It is highly informative to add information on localization of toxin genes (plasmid or chromosome) in all strains.

  1. P15L468 “All botulinum neurotoxin genes” should be “All botulinum neurotoxin sub-type B1 genes”.

  1. P16L511~ L517 This part is difficult to understand.

  1. TableS2 C. botulinum strain 111 possesses bont/X gene and orfX genes on a chromosome, and bont/B2 gene and ha genes on a plasmid.

Author Response

Authors response: Thank you for the thorough review, positive feedback and helpful suggestions.

P1L26 Names of bacteria should be in italics.

Authors response: The requested change has been made

P2L77 Please add a definition of “B5F2”, “A5B3”…, notations of toxin types in two large characters.

Authors response: Text has been added to the introduction and methods sections to clarify this point.

P3L102 C. botulinum strain ATCC3502 is also known as strain Hall. It would be helpful to include this information.

Authors response: The requested change has been made

P3L117 Fig.1 Please add explanation of abbreviations, “N”, “I”, “F”, “W”, “U” in legend.

Authors response: The requested change has been made

P6 Fig.2 Please add explanation of the right side black column “new” in legend.

Authors response: The requested change has been made

P9L261 Please include enough information about “variant group” of MLST.

Authors response: The strains could not be assigned to existing MLST STs. Full MLST data are in Table S2, with an overview presented in the manuscript.

It is highly informative to add information on localization of toxin genes (plasmid or chromosome) in all strains.

Authors response: We agree that this information is informative, however this has not been done as the genome sequences don't allow us to conclusively state whether plasmid or chromosomal, but only whether surrounding genes are commonly plasmid- or chromosome-associated.

P15L468 “All botulinum neurotoxin genes” should be “All botulinum neurotoxin sub-type B1 genes”.

Authors response: The requested change has been made.

P16L511~ L517 This part is difficult to understand.

Authors response: The text has been reworded to avoid confusion.

TableS2 C. botulinum strain 111 possesses bont/X gene and orfX genes on a chromosome, and bont/B2 gene and ha genes on a plasmid.

Authors response: The requested correction has been made.